# Drivers of Routine and Outbreak Vaccination Uptake in the Western Democratic Republic of Congo: An Exploratory Study in Ten Health Zones

**DOI:** 10.3390/vaccines10071066

**Published:** 2022-07-01

**Authors:** Laurene Peckeu-Abboud, Patrick Mangoni, Kaouther Chammam, Papy Kwete, Patrick Mutombo Lupola, Veerle Vanlerberghe, Jepsy Yango, Marie Meudec, Christian Ifufa, Marianne A. B. van der Sande, Joule Madinga Ntwan, Placide Mbala

**Affiliations:** 1Institute of Tropical Medicine (ITM), 2000 Antwerp, Belgium; lpeckeu@itg.be (L.P.-A.); kchamman@itg.be (K.C.); vvanlerberghe@itg.be (V.V.); mmeudec@itg.be (M.M.); 2Institut National de Recherche Biomédicale (INRB), Kinshasa 1197, Congo; patrick.mangoni@unikin.ac.cd (P.M.); jfkfirst@gmail.com (P.K.); patrickmutombo.inrb@gmail.com (P.M.L.); jepsyango@gmail.com (J.Y.); ifufachristian@gmail.com (C.I.); joulemadinga@gmail.com (J.M.N.); placidembala@inrb.net (P.M.); 3Julius Centre for Health Sciences and Primary Care, UMCU, 3584 CG Utrecht, The Netherlands

**Keywords:** vaccination uptake, outbreaks, DRC, HCWs

## Abstract

We performed a cross-sectional survey on vaccination-related knowledge, attitudes, and practices (KAP) among randomly selected parents of <5 years-old children, elderly populations (aged ≥ 55 years), and health care workers (HCWs) in 10 health zones from 4 provinces of the Democratic Republic of Congo (DRC). Questionnaires targeted both routine (BCG, measles, polio) and outbreak-related (cholera, Ebola, COVID-19) vaccinations. In total, 2751 participants were included, 1165 parents, 1040 elderly, and 546 HCWs. In general, KAP expressed were supportive of vaccination uptake, although concerns regarding side effects and feelings of being insufficiently informed were more prevalent among parents and the elderly. Vaccine acceptance was lower for outbreak vaccinations (57%) than for routine vaccinations (90%). HCWs expressed the highest vaccine acceptance. Problems with the vaccine supply chain were reported by 20% of respondents. Despite a high level of positive KAP towards vaccination, parents and the elderly expressed a need to be better informed and had concerns regarding vaccine side-effects. A high acceptance for routine vaccinations was reported by participants, but somewhat less for outbreak vaccinations. In conclusion, HCWs in the communities could play a key role in the increased uptake of routine vaccinations and in optimizing uptake during outbreaks, provided that the supply chain is functioning well.

## 1. Introduction

Vaccinations are one of the main public health achievements of the previous century. Timely vaccinations using safe and effective vaccines have safeguarded millions of children and adults worldwide from disease, disability, and death [1]. They are central to protect public health from infectious diseases. Vaccinations are thereby one of the most cost-effective public health interventions available (https://data.unicef.org/topic/child-health/immunization; accessed on 8 February 2022). The vaccination of key populations or of general populations is also central to many outbreak interventions.

Despite the establishment of national immunization programs, currently operational in every country in the world, vaccination uptake is often suboptimal [2]. This can be related to the insufficient demand for vaccinations. This demand might be reduced among intended recipients following peer or political pressure, or related to more individual factors such as lack of knowledge, concerns about potential short- or long-term side effects after vaccination, different perceptions concerning disease severity and prevention, poor experiences following previous vaccinations, or encounters with health services [3]. On the other hand, “supply chain” factors, such as insufficient access to affordable vaccines, substandard quality due to insufficient quality controlled production, transportation and storage, or factors related to health service delivery, such as the disruption of regular health services due to external factors, lack of availability of quality vaccination services, or the lack of support from health care providers for specific vaccinations, can also impede vaccination uptake.

Furthermore, major infectious disease outbreaks such as cholera, Ebola, or COVID-19 (and other emergencies) tend to disrupt routine health services, and can have a detrimental effect on the utilization of routine health services, including routine childhood vaccination [4,5,6,7,8]. When and where this disruption happens, a large group of young children remains susceptible to vaccine preventable diseases (VPDs). This causes a high risk of subsequent outbreaks, aggravating the health and societal impact of a primary outbreak. Introduction of novel (outbreak-related) vaccinations can further complicate vaccine uptake. During the first wave of the current COVID-19 pandemic, more than half of all countries reported interruptions of routine vaccination services, putting millions at risk [9]. Some countries reported a greater than 50% lower vaccine uptake in 2020 compared to 2019 [10,11]. The ongoing COVID-19 pandemic and the resulting vaccination campaign efforts, has illustrated the many challenges related to widespread vaccination in sub-Saharan Africa [12,13,14,15,16]. 

The Democratic Republic of Congo (DRC) has been prone to outbreaks of multiple VPDs, such as measles, vaccine-derived polio, cholera, meningitis, and Ebola [17]. Often, dedicated vaccination campaigns have been initiated to curb ongoing transmission and stop such outbreaks. However, during previous Ebola outbreaks, the diversion of routine vaccination efforts due to outbreak vaccination campaigns resulted in a drop in measles vaccination uptake, with consecutive measles outbreaks in the Ebola outbreak region aggravating the situation [18].

Nevertheless, based on an analysis of surveillance data, the uptake of routine vaccination in Kinshasa, the capital of the DRC, had not decreased in the first six months of the COVID-19 pandemic [19]. At the same time, a 2020 cross-sectional study among DRC adults observed a willingness to vaccinate against COVID-19 among only 56% of participants, and the percentage was even lower among the subset of health care workers (HCWs) (28%) [20,21]. Thus, to enable optimal vaccination impact, it is crucial to have a clear understanding of the determinants of insufficient uptake, both on the demand side (e.g., knowledge, perception, vaccine acceptance, socio-economic factors) and on the supply chain side (e.g., resources, cold chain) by time, place and person.

In this study, we aimed to identify promoting and hampering determinants of routine and outbreak vaccination uptake among different stakeholders (parents of <5 years-old children, the elderly population (aged ≥55 years), and HCWs) in different geographical settings.

## 2. Methods

### 2.1. Study Sites and Populations

The study was conducted in four western provinces of the DRC: Kinshasa, Kongo Central, Kwango, and Kwilu. The DRC is administratively divided into 26 provinces and each province is divided into health zones (HZ), 517 in total. We selected 10 HZs, aiming for representative samples of these 4 provinces, taking accessibility of the areas into account and at the same time, we selected half of the HZ in rural, and half in urban areas. An HZ is the operational unit of the health system in the DRC, which includes around 100,000 inhabitants, with 1 referral hospital (Hôpital Général de Reference, HGR), 1 central office (Bureau Central de Zone, BCZ) and several health centers (Centre de Santé, CS) and health posts (Poste de Santé, PS). Each HZ is divided in health areas (HA), which contains a dozen villages in rural settings, and streets in urban settings. Each health area contains at least one health center, which is in charge of providing primary health services.

In each selected HZ, we included three different populations: (i) parents of <5 years-old children (the target group for routine early life vaccinations); (ii) the elderly population (aged ≥ 55 years, one of the target groups for COVID-19 vaccinations in the DRC, and potentially, for other future vaccinations); and (iii) HCWs, including community health workers (both at risk due to high exposure, and key informants in offering vaccinations to eligible populations). Participant groups were mutually exclusive. To be eligible, participants had to have been residents in the area for at least 6 months.

We aimed to randomly include in each HZ at least 20 HCWs, 50 parents, and 50 elderly citizens, which would allow us to identify factors reported by at least 20% of each population group with a precision of ±6% for HCWs and ±3.5% for the other two groups (using command cii prop, Stata Corp vs. 15).

For the selection of parents and elderly populations, a four-stage cluster sampling method was used to select health areas, villages (or cities), households, and participants within the households. Households were selected following the EPI household sampling scheme, developed by the World Health Organization’s Expanded Program on Immunization (EPI) [22]. The EPI method was used until the predetermined sample size for each group (50 parents of <5 years-old children and 50 elderly) per HZ was reached.

Stratified selection of HCWs was performed. First, in a selected HZ, the HGR and BCZ were systematically visited, and in addition, 4 PS in 4 CS were randomly selected. Three wards in the HGR were randomly selected, and 1 HCW was randomly selected in each included ward. For each HZ, 3 HCWs of the BCZ were randomly selected. The remaining HCWs were randomly selected in each health center. HCWs who were part of these structures were included until the predetermined sample size of 20 HCWS per HZ was reached.

### 2.2. Data Collection

We conducted a cross-sectional study using previously validated questionnaires [23,24,25,26]. Data were collected on demographics, and on six knowledge, attitudes, and practices (KAP) concepts (belief, knowledge, attitude, risk perception of side effects, feeling informed towards vaccination, and risk perception of VPDs). An additional KAP concept of trust (combining trust towards vaccination, vaccination programs, and providers) was investigated only among the parents of <5 years-old children. Vaccine acceptance was assessed for specific routine (Bacillus Calmette-Guérin (BCG), measles, and poliomyelitis) and outbreak (cholera, Ebola, COVID-19) vaccines. In the HCW population, we also collected data on the supply chain of vaccines.

A pilot study was conducted in both rural and urban HZ to test and fine tune the questionnaire, the selection of participants, verbal consent seeking, and the data collection tools. Following the pilot study, 139 questions divided in 5 sections were included in the questionnaire. Questions regarding KAP concept, vaccine acceptance, and supply chain analyses are detailed in a Appendix A.

In each HZ included in the study, local field workers were recruited and trained in study procedures. Questionnaires were administered in French or translated into local languages, where appropriate. Data were collected on mobile tablets using KoBo Toolbox. No personal identifiers were collected, except for the area where the interview took place, as well as the sex and age category. Administrative permission was obtained from each provincial authority prior to roll out of the survey per province, as well as from the authorities in the selected HZs.

Data were collected between June and September 2021. During this time, COVID-19 vaccination became available in DRC on a very limited scale only, and was not (yet) offered in any of the included HZs during the time of the survey.

### 2.3. Statistical Analysis

Descriptive analyses were performed to characterize the study populations and the vaccination supply-related issues reported by the HCWs. Median and interquartile ranges (IQR) were estimated for non-normally distributed variables. Frequencies were described for categorical variables.

For questions related to the six KAP concepts, we used a 3-point Likert scale. Based on this scale and the number of questions included in a particular KAP concept, a respondents’ score (per KAP concept) was computed by summing the Likert scale points (1 = disagree, 2 = neutral, and 3 = agree) and dividing them by the number of questions included for the KAP concept.

To illustrate this, the KAP concept of belief is based on 4 questions:E.g.: score of belief = (q1:agree = 3) + (q2:agree = 3) + (q3:neutral = 2) + (q4:neutral = 2)

Note that the computation of the score of risk perception on VPDs was based on one question, “If I do not vaccinate, then the risk that I will get and transmit one of these infectious diseases is…” Here, a 4-point Likert scale was used, coded as followed: 1 = the risk is weak, 2 = the risk is equal (whether I vaccinate or not), 3 = I don’t know, and 4 = the risk is high.

We then explored whether the six KAP concepts varied by urbanization status (rural versus urban areas) and among group of participants (parents, elderly, HCWs). Wilcoxon and Kruskal–Wallis tests were used, respectively. A post hoc analysis (Dunn’s multiple comparison test with Benjamini–Hochberg adjustment) was used when significant differences between the participants’ groups were found. For the concept of trust, since only parents were interrogated, the comparison between rural and urban areas was performed using the Wilcoxon test.

We also compared vaccine acceptance (defined as the degree to which individuals accept, question, or refuse vaccination [27]) between rural and urban areas, group of participants (parents, elderly, HCWs), KAP concepts, and awareness (or not) of recent cases in regard to the outbreak diseases investigated (cholera, Ebola, COVID-19). In this analysis, each KAP concept was split into 3 levels: negative (score < 2), neutral (score = 2), and positive (score > 2) values.

For the risk perception of VPDs, the levels were: perception of the risk of disease as low (score < 3), neutral perception of the risk of disease (score = 3), and perception of the risk of disease as high (score > 3). The Kruskal–Wallis test and post hoc analysis (Dunn’s multiple comparison test with Benjamini–Hochberg adjustment) were also used.

All analyses were performed using R version 3.6.3. Scripts are accessible via the GitHub repository: https://github.com/Laureneitm/fivacc_analysis accessed on 8 February 2022.

## 3. Results

### 3.1. Study Population

In total, 2771 persons were visited and 2751 participants consented to take part of the study, of which 1165 (42%) were included as parents, 1040 (38%) as elderly citizens, and 546 (20%) as HCWs. The majority of participants (1792; 65%) were female; the median age was 30 (Interquartile Range (IQR): 25–37) years for parents, 59 (IQR: 57–65) years for the elderly, and 41 (IQR: 33–51) years for HCWs. The number of participants varied from 214 to 413 between the 10 study areas; 1297 (47%) of the participants were from rural areas. Nearly half of the participants (1282, 47%) had achieved secondary school level and 625 (23%) went to university (Table 1).

### 3.2. Analysis of the KAP Concepts

Overall, in both urban and rural areas, the scores for the six KAP concepts were high (Figure 1), reflected in general positive beliefs and attitudes, adequate knowledge, and risk perception on side effects towards vaccination, adequate risk perception on VPDs, and a perception of being sufficiently informed.

However, significant differences between rural and urban areas were found for the score of belief (Wilcoxon-test *p*-value = 0.01) and knowledge (Wilcoxon-test *p*-value = 0.002) (Figure 1a,b). The risk of developing side effects after vaccination was perceived as rarer by rural populations compared to the urban population (Wilcoxon-test *p*-value < 0.001) (Figure 1d). The risk of contracting VPDs when not vaccinated was perceived as lower in urban areas (Wilcoxon-test *p*-value < 0.001) (Figure 1f). Figure 2 depicts the distribution of the scores of the six KAP concepts per group of participants. No difference was reported among these groups for the KAP concepts of belief, knowledge, risk perception regarding VPDs, and attitude, but the elderly population perceived a higher risk of developing side effects after vaccination compared to HCWs (post hoc Dunn test *p*-value = 0.002) (Figure 2d). Parents and elderly populations felt less informed to decide on the uptake of vaccinations compared to the HCWs (post hoc Dunn test *p*-value < 0.001) (Figure 2e).

For parents, the concept of trust was compared between rural and urban areas. Overall, the level of trust towards vaccination, vaccination programs, and providers was high for the majority (the maximal score reached 85%). Parents living in rural areas reported a higher level of trust than parents living in urban areas (Wilcoxon-test *p*-value = 0.04).

### 3.3. Vaccine Acceptance

Vaccine acceptance was high for routine vaccinations, with 90% of the participants fully accepting all routine vaccinations (Figure 3a). Taking the routine vaccinations independently from one another, 96%, 93%, and 96% of participants fully accepted BCG, measles, and polio vaccinations, respectively. In contrast, vaccine acceptance for outbreak vaccinations was lower: 57% would accept the 3 different vaccinations (Figure 3b). Vaccine acceptance was lowest for COVID-19 (66%), compared to Ebola (78%), and cholera (80%).

Since vaccine acceptance for outbreak vaccinations was heterogeneous, we stratified our analyzis by rural and urban areas, by group of participants, and by level of KAP concept (Figure 4). Vaccine acceptance during outbreaks was higher in rural areas (Wilcoxon-test *p*-value < 0.001), higher among HCWs (post hoc Dunn test *p*-value < 0.001), higher among people with positive beliefs and attitude towards vaccination (post hoc Dunn test *p*-value < 0.001 for both concepts), and higher among people who perceived the risk of developing side effects after vaccination as low (post hoc Dunn test *p*-value = 0.006 compared to people who perceived the risk of developing side effects as high, and *p*-value < 0.001, compared to people who perceived the risk of developing side effects as neutral). Vaccine acceptance for outbreak vaccination was lower among people with incorrect knowledge (post hoc Dunn test *p*-value < 0.001) or people feeling insufficiently informed on vaccination (post hoc Dunn test *p*-value = 0.002, compared to people with neutral feelings, and *p*-value < 0.001, compared to people feeling sufficiently informed). Vaccine acceptance was also lower among people with a neutral risk perception of the outbreak (post hoc Dunn test *p*-value < 0.001). 

Awareness of people who had been affected by an outbreak was associated with higher acceptance of that specific outbreak vaccine (Wilcoxon-test *p*-value < 0.001 for the 3 specific vaccines) (Figure 5).

### 3.4. Supply Chain Analysis

A total of 109 HCWs (20%) reported to have experienced interruption in the supply chain of vaccines in the preceding year. Most common problem reported were related to unavailability of vaccines (12%), closely followed by cold chain interruptions (12%); 8% reported problems with the availability of other materials (such as syringes).

## 4. Discussion

The DRC is an epidemic-prone country with limited resources, but considerable experience in outbreak control. In 2020, the first year of the COVID-19 pandemic, the DRC also experienced a new Ebola outbreak in Equateur province, just after the large 2018–2020 Ebola outbreak in Kivu came to an end, and the cholera outbreak at that time in the Congo River Basin was the largest in the region. In this context, we observed high acceptance for routine vaccinations, but a lower acceptance with regard to outbreak vaccinations. As we included 3 different target groups from both urban and rural districts, this is likely to be a representative reflection of KAPs towards (outbreak) vaccinations.

Comparison of routine acceptance versus COVID-19 acceptance was previously performed in 5 West African countries (Burkina Faso, Guinea, Mali, Senegal, and Sierra Leone). In this study, concerns about COVID-19 side effects and believing that COVID-19 vaccines carry more risk than routine vaccines were reported as lowering the willingness to get vaccinated [28]. In a focus group discussion study in Zambia, acceptance of vaccines in general was high for all participants, because vaccinations were associated with child health [29]. These results could partially explain why communities accept more routine vaccinations compared to outbreak vaccinations, especially COVID-19 vaccines.

We found that HCWs generally had higher vaccine acceptance for outbreak vaccinations. This is in line with a recent review, where 13 studies (out of the 15 selected) observed higher acceptance of vaccines among HCWs [30]. High vaccine acceptance in HCWs provides a key opportunity to involve them early and comprehensively in future campaigns. For many people, the first line health care provider is the person they most trust to provide them with relevant and sufficient information. At the same time, for HCWs to fulfill this role effectively and credibly, it is key that the supply chain is functioning well, to enable people to translate vaccine acceptance in the actual timely uptake of vaccination with a quality-assured vaccine. The high acceptance among HCWs to vaccinate during outbreaks is also reassuring, as this may help curb nosocomial transmission in such settings. A recent seroepidemiologic study among HCWs and their families in Kinshasa, DRC, observed that transmission was more prevalent in communities than in health centers (Mandinga et al., submittted). In regard to outbreak vaccine acceptance, we also observed that awareness of people who had been directly affected by an outbreak was associated with higher outbreak vaccine acceptance. It was previously described elsewhere that disease susceptibility perception can influence vaccine acceptance, whereby vaccine acceptance is higher in groups who perceive vaccination as an important entity to counter the detrimental consequences of VPDs [31].

It is noteworthy that in a DRC survey a year before ours, overall acceptance of COVID-19 vaccination was lower (56% then vs. 66% here), while acceptance among HCWs was lower rather than higher, compared to others (28%) [20,21]. Although these are different surveys, with different methodologies, and thus not comparable, a higher acceptance in 2021 might be related to the ongoing pandemic in the year between the surveys, or it might be related to study differences, such as different provinces and study populations included or a different methodology used. In particular, as the 2020 survey was online, self-selection of participants may have occurred, whereby they may also have been exposed more to rumors and ‘fake news’ via other online sources.

Living in rural areas was identified as a protective factor associated with having children immunized in 24 African countries, including the DRC [32]. In our study, we observed that rural populations had positive beliefs and better knowledge towards vaccination, while the risk of developing side effects was perceived as rarer compared to urban populations. Our results are in line with a 2015 study in rural DRC on drivers of routine vaccination uptake and point to the essential need of better communication and education on vaccines for urban populations, where access to vaccination centers and the relatively higher level of education can provide multiple opportunities to trigger the subsequent use of immunization services [33]. Communication and educational campaigns could also help in limiting the spread of rumors and distrust among urban areas, where these can circulate more rapidly and reach a higher number of people than in other settings.

Parents and the elderly felt less informed on vaccination and perceived a higher risk of developing side effect than HCWs. As fear of side effects fuels vaccine hesitancy [29], it is crucial to better inform these populations. This is also important, since elderly populations could play an even bigger role than HCWs in influencing the decisions of younger family and communities [34,35].

Attitudes, beliefs, risk perceptions, feeling of being sufficiently informed toward vaccination, and vaccine acceptance are not static. They can differ between populations and places, and they change over time. This information is key to enable agile, targeted interventions by public authorities to optimize the uptake of vaccinations when aiming to prevent and control outbreaks. As discussed earlier, our study clearly demonstrated differences by place and population group, but logistical challenges to implement such a sentinel survey made it impractical to conduct follow-up rounds. Alternative solutions could be explored, by, e.g., including repeated assessment as part of routine services (accounting for bias due to health care seeking populations), or by establishing an open web-based cohort to collect such data (accounting for self-selection bias). Moreover, our study was not designed to explore in depth the potential mechanisms behind expressed concerns, such as fear of side effects, or differences in intended uptake between rural and urban populations. Now that we have been able to elucidate such critical concerns, follow-up studies are warranted to better understand these outcomes.

The strength of our study is the use of using previously validated methodology to assess KAP concepts and vaccine acceptance, adapted to the DRC setting, as well as the timeliness of the study during the gradual spreading of the COVID-19 outbreak across the country. In addition, we were able to include a range of populations coming from different settings. In spite of significant administrative and logistical challenges, all HZs could be surveyed within a time period of 4 months. While our study is one of very few large-scale studies assessing KAP concepts and vaccine acceptance in the DRC, some limitations have to be recognized. The first limitation is the cross-sectional design, which did not allow assessing and exploring drivers of trends in KAP concepts and vaccine acceptance. In addition, we used a 3-point Likert scale to assess KAP concepts and vaccine acceptance. In the psychometric field, it is commonly accepted that when measuring a construct that falls on a continuum from low to high (such as KAP concepts and vaccine acceptance), more reliable (consistent responses) and valid (reflects true attitudes and opinions) results will be obtained when the rating scales include more points [36]. However, when many items are investigated, as in our study, it has been shown that a 3-point scale does not diminish the reliability nor validity of the resulting scores [37], especially when scores are averaged across people and across many items [38]. Assessing opinions and behaviors using quantitative methodology could mask nuances in opinions. Qualitative data could have improved our understanding of KAP and vaccine acceptance, especially when people had a lower confidence regarding vaccination; mixed method studies to explore this further are ongoing.

In conclusion, different DRC populations showed high acceptance for routine vaccinations, but somewhat less for outbreak vaccinations, especially if there was no direct experience with the outbreak. Though a high level of positive KAP was reported towards vaccination, parents and the elderly expressed their desires to be better informed and concerns regarding vaccine side-effects. Implementation of targeted interventions can positively influence vaccination uptake. HCWs are often the most trusted source for health-related information and had a higher acceptance to uptake, but also reported frequent interruptions in the vaccine supply chain. Although we did not directly assess the willingness of HCWs to act as promotors of uptake, it is likely that, provided that supply constraints can be addressed, HCWs can be major agents of change to increase the uptake of live-saving vaccines.

## Figures and Tables

**Figure 1 vaccines-10-01066-f001:**
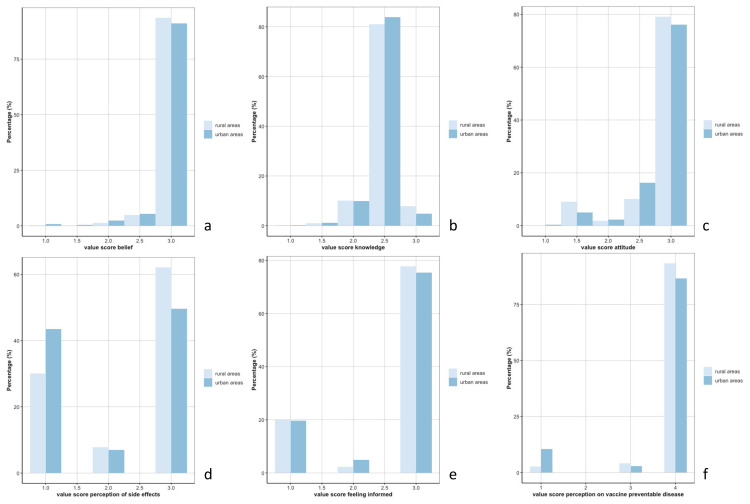
Distribution of the scores for the 6 concepts—(**a**): belief, (**b**): knowledge, (**c**): attitude, (**d**): risk perception of side effects, (**e**): feeling informed, (**f**): risk perception on vaccine preventable diseases—between rural and urban areas.

**Figure 2 vaccines-10-01066-f002:**
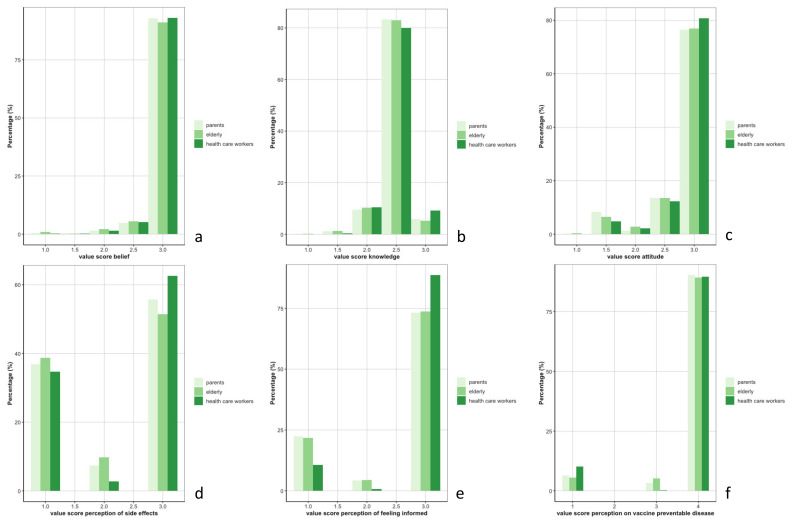
Distribution of the scores for the 6 concepts—(**a**): belief, (**b**): knowledge, (**c**): attitude, (**d**): risk perception of side effects, (**e**): feeling informed, (**f**): risk perception on vaccine preventable diseases—among group of participants (parents, elderly, and health care workers).

**Figure 3 vaccines-10-01066-f003:**
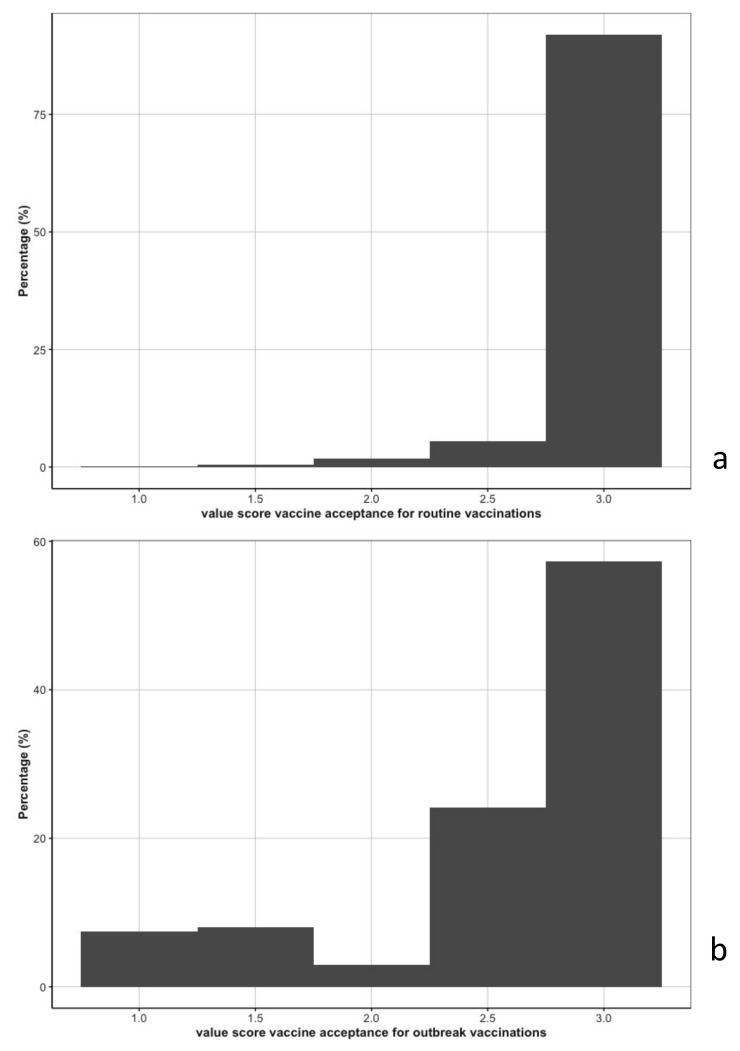
Vaccine acceptance distribution for routine vaccinations (3**a**) and outbreak vaccinations (3**b**).

**Figure 4 vaccines-10-01066-f004:**
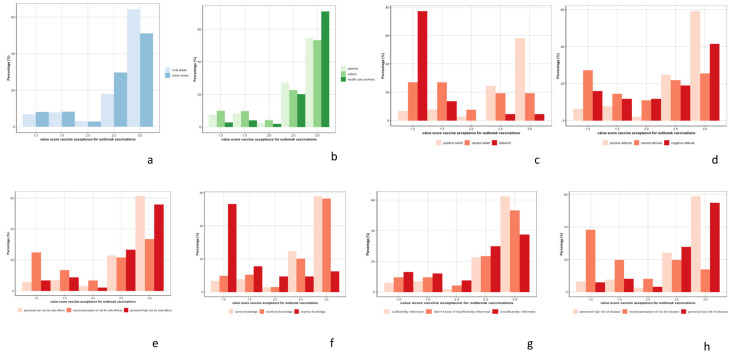
Distribution of vaccine acceptance for outbreak vaccinations between rural and urban areas (**a**) by group of participants, (**b**) parents, elderly, and health care workers; and level of concepts, (**c**) belief, (**d**) attitude, (**e**) risk perception of side effects, (**f**) knowledge, (**g**) feeling informed, (**h**) risk perception on vaccine preventable diseases.

**Figure 5 vaccines-10-01066-f005:**
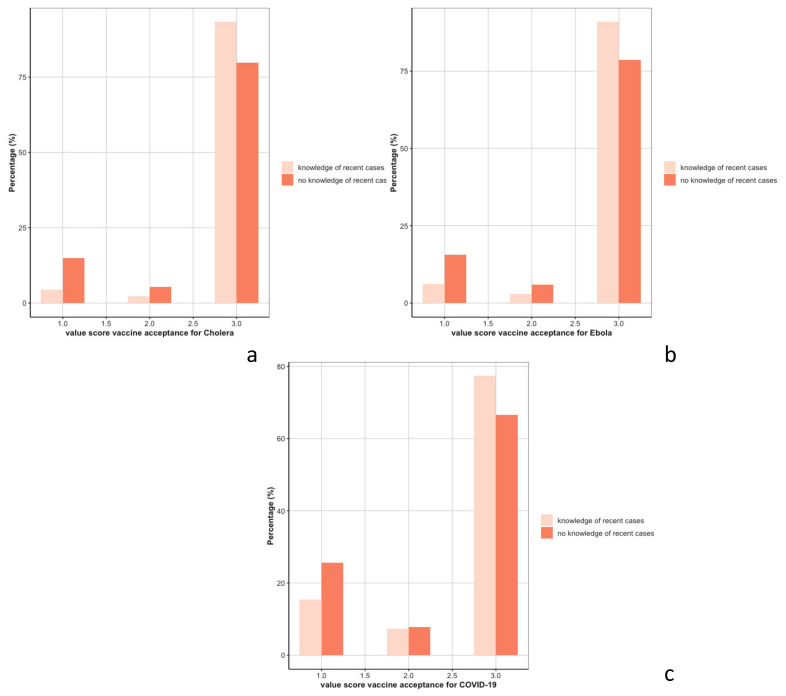
Distribution of vaccine acceptance for cholera (**a**), Ebola (**b**), and COVID-19 vaccines (**c**) by awareness of recent cases.

**Table 1 vaccines-10-01066-t001:** Characteristics of participants included in the study (n = 2751).

Participant Characteristics	All Participants	
	(n = 2751)	
**Health zones**	**n (%)**	
Boko	280 (10)	
Boma	249 (9)	
Kenge	226 (8)	
Kikwit nord	306 (11)	
Limete	260 (9)	
Lusanga	276 (10)	
Masi Manimba	296 (11)	
Matadi	413 (15)	
Mbanza Ngungu	231 (8)	
Nsele	214 (8)	
**Gender**	**n (%)**	
Female	1792 (65)	
Male	959 (35)	
**Participant group**	**n (%)**	**Age (in years) (median, IQR)**
Health care workers	546 (20)	41 (33–51)
Parents of children < 5 years old	1165 (42)	30 (25–37)
Elderly (≥55 years)	1040 (38)	59 (57–65)
**Educational level**	**n (%)**	
None	225 (8)	
Primary school	581 (21)	
Secondary school	1282 (47)	
University	625 (23)	
Other	25 (1)	
Don’t know	9 (<1)	
Refused to respond	4 (<1)	

## Data Availability

Available online: https://github.com/Laureneitm/fivacc_analysis.

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
