# Peer review of "Drivers of Routine and Outbreak Vaccination Uptake in the Western Democratic Republic of Congo: An Exploratory Study in Ten Health Zones"

_vaccines, 2022, doi:10.3390/vaccines10071066_

Round 1
Reviewer 1 Report
The research is essential for stakeholders, not just for routine vaccination but also for emerging diseases. We should thank the authors for their efforts. However, I have some observations for the analysis to improve the soundness of their results and conclusions.
Methods.
1) Authors randomly included different groups to compare the KAP differences. It is unclear if there were statistical differences between sociodemographic characteristics in selected and not selected persons. That will be important to identify a selection bias, and if researchers detect will be necessary to describe how to address it.
2) Analysis. We recommend reporting 95%CI for each percentage or prevalence.
3) Its recommended, if possible to estimate probabilistic weights for using the SVY command in stata analysis.
4)Its recommended, if possible the use of some multivariate regression method for the comparison to estimate the difference between groups and rural and urban.
Results. 2,751 participants is a large sample because the authors are analyzing bivariate analysis only. So the p-values could be spurious because of a big sample.
Author Response
Thank you for your encouraging feedback and suggestions!
Ad 1: Participants were included following a random sampling scheme, but not selected from a given population. As such, we collected no data from other people, not included in the sample.
Ad 2: As data were not normally distributed, we have expressed results in median values with IQR.
Ad 3: We would gladly do this is we had used Stata for analysis, but data were analysed with R 3.6.3, and we did not use the Stata syv command. The link to the scripts is provided in the paper. We only used Stata for the sample size calculations.
Ad 4: We agree, but due to heterogeneities, we decided it was more appropriate to provide stratified rather than multivariable analysis.
Ad Results: We agree, but as we needed to use non-parametric tests, this was the best method to provide results of significant differences, which may not be obvious from eyeballing.
Reviewer 2 Report
I have reviewed the manuscript “Drivers of routine and outbreak vaccination uptake in the western Democratic Republic of Congo: an exploratory study in ten health zones” submitted to “Vaccines” for publication. In this study, the authors have conducted a cross-sectional survey on vaccination-related Knowledge, Attitudes and Practices (KAP) among randomly selected parents of <5 years-old children, elderly populations (aged 55 years) and health care workers (HCWs), in 10 health zones from 4 provinces of Democratic Republic of Congo. This manuscript fits well within the scope of the journal; it needs some improvements; there are a few suggestions that authors may consider to improve it further:
The use of English language is reasonable, however, there are a number of punctuation and grammatical errors; that should be corrected and rephrased using academic English for a better flow of text for the reader.
- Abstract is appropriate, however the findings are not clear, some of the results can be summarized in the abstract.
Reference citations should be corrected as citations should not be superscripted.
Introduction 1st paragraph, please cite appropriate references where possible; the following recent article can benefit by supporting several statements
https://www.mdpi.com/1660-4601/18/21/11008
Otherwise, the introduction; is detailed, and compact, covering the background information and the rationale of the study effectively.
Result data and Figures are well presented and described in the text
The discussion section is well-written and the results are discussed comprehensively.
Author Response
Thank you for your positive feedback!
Re your specific suggestions:
-we have had the manuscript checked by a native english speaker
-to have the results stand out more clearly in the abstract, we have separated the text in paragraphs, which may clarify that over half of the text is dedicated to a summary of main results.
-we apologise, and have now corrected the references to the format requested
-we have added a reference to the first para of the introduction, which summarised the large impact of vaccinations on public health, as requested

Reviewer 3 Report
This is an interesting and important study that fills a gap in the existing literature with reference to a key issue such as differences in vaccine acceptance between different types of vaccinations in a large African country such as DRC. The information provided in the paper is certainly valuable, but a few important revisions are in order for the study to be publishable. I will detail them below. In the computation of risk perception, the choice of coding ‘I don0t know’ with 3 and ‘risk is high’ with 4 is pretty awkward and can be a source of bias. Please discuss and motivate this choice. What are possible explanations for differences in trust and vaccine acceptance between urban and rural areas? A discussion of this important point should be provided, possibly supported by appropriate literature. Likewise for the differences between routine and outbreak related vaccinations, and in particular for the low acceptance of COVID-19 vaccines. Are the two DRC surveys on COVID-19 vaccine acceptance really comparable? If so, please explain more clearly why. If not, please avoid making misleading comparisons. The study design does not really take into account infodemics and misinformation as a possible source of vaccine hesitancy in the population. The authors should discuss and motivate this choice, as COVID-19 related misinformation is likely to explain at least in part the higher vaccine hesitancy with respect to other vaccines. On the other hand, the authors mention misinformation in their discussion. One really wonders why the study design did not focus more on this issue. The fact that rural populations are better informed and more positive about vaccination is interesting and also a bit surprising, as it runs counter to what is commonly found in other countries. This result too should be discussed and interpreted more in depth. Fear of side effects seems to be an important factor for vaccine hesitancy in the study population. Where does this fear come from? What kind of information sources mostly account for such attitude? This is an important question to answer in a study like this, and if the issue has not been considered in the design, this is a serious limitation that should be discussed and acknowledged. The study places a high emphasis on the fact that since HCW are more positive toward vaccination than the other populations, they could play a bigger role in future campaigns. While this is intuitively reasonable, to be precise the study only measures HCW attitudes toward their own vaccinations, and not their willingness to act as testimonials of vaccination toward other people. While this could be the case, it is however, given the study design, an arbitrary deduction that is not supported by the collective evidence, and as such the policy implications that are suggested in the study are not empirically supported. The authors should be aware of this in revising the paper. There are minor language errors and inaccuracies that should be carefully checked.Author Response
Dear reviewer,
Thank you for your valuable comments and suggestions, which we tried to address as follows:
-coding of answers was based on coding in comparable validated surveys. However, we agree that this weighting can be context-dependent, and that other weightings could appropriate also. We did not have the means to test nor validate this however, which would also have gone beyond the scope of the current study.
-we regret that this study was not designed to identify drivers of different outcomes between areas (or respondents), which is why we presented stratified analyses. As this reviewer made additional requests for a more qualitative in-depth study around barriers (see below), we have clarified the scope of our study, and the usefulness to explore in more depth the barriers now identified.
-we agree the DRC studies are not directly comparable, and apologise if we did not clarify this. We have modified the text to make this more clear.
-indeed, the objective of this study was to elucidate the role of infodemics and misinformation on vaccination uptake, as this would have required a different mixed method design (such studies have been initiated by our team also). We would like to point out that acceptance for all outbreak vaccinations, was lower than for the routine vaccinations, whereby intended uptake of Covid19 vaccination was included as one of the three outbreak vaccinations surveyed.
-we agree with the reviewer that in-depth qualitative data could explore additional questions related to vaccination uptake (such as origins of fear of side effects, but also differences urban-rural), but this was beyond the scope of this study. In fact, we suggest that it might be more useful to do such studies now that we have identified such drivers, rather than if we had planned a priori to explore in depth potential drivers for selected assumed barriers, which may not have materialised. As requested, we have added this limitation to the discussion.
-we agree and have now acknowledged that we did not collect data on the willingness of HCWs to encourage uptake among their patients.
Round 2
Reviewer 2 Report
Many thanks for revising the manuscript.